## Research Article

Community-led projects; Nature restoration; Biodiversity loss; Multi-stakeholder approach; Participatory Evaluation

**Corresponding author:**
Emma Verling;
Email: emma.verling@ucc.ie

‡This article has been updated since its original publication. A notice detailing this can be found here: https://doi.org/10.1017/cft.2025.10007

# The power of learning from the bottom up: working towards a blueprint for community-led biodiversity protection and restoration‡

Emma Verling[1] , Maria Power[2], Melanie Biausque[1,3], Lee Wah Pay[4], Aoife Deane[1], Rory Scarrott[1] and Darragh Ó Súilleabháin[4]

[1]MaREI, the SFI Research Centre for Energy, Climate and Marine Environmental Research Institute, University College Cork, Cork, Ireland; [2]Community Consultants, Waterford, Ireland; [3]British Geological Survey, Belfast, UK and [4]Cork County Council, Cork, Ireland

## Abstract

The impacts of climate change have become more widespread and frequent, and society is beginning to recognise the connection between it and the biodiversity crisis. Communities have the capacity to play a key role in the success of multi-stakeholder nature restoration projects, but examples of successful projects, in which communities are the architects of the action – as opposed to the recipients of it – are not well documented. This study used a participatory evaluation research approach to explore a multi-stakeholder, community-led restoration project at Harper's Island Wetlands, Co. Cork, Ireland to understand the elements of success and to extract key learnings for other communities. In order to rapidly upscale nature restoration and biodiversity protection globally, there is an urgent need to gain speed and momentum, identifying innovative approaches and disseminating them appropriately. The insights from this case study highlight four key components to be considered by groups at the beginning of community-led projects: setting up a core committee, assigning clear roles within the committee, creating a short-, medium- and long-term strategy and beginning practical tasks as soon as possible. This research serves as a step towards preparing blueprints to inform research, policy and practice in this space to enable stakeholders to respond collectively

## Impact Statement

We illustrate and describe the development of a successful community-led nature restoration project which is supported by a local authority and an NGO. This work seeks to highlight the power and benefit of supporting communities in activities related to biodiversity protection, nature restoration and climate action. We determine that our research reveals a number of important elements of success which will be key to consider for broader upscaling and replication of restoration activities. The research offers unique perspectives about the varied motivations of communities, which can sometimes be viewed as a barrier to progress. In fact, these motivations, while rooted in local contexts, can serve to strengthen restoration and biodiversity protection if they are supported appropriately. Acknowledging and supporting the things that motivate individuals and communities can lead to powerful outcomes that can, in turn, address global motivations such as climate action. There is an urgent need for increased restoration efforts across Europe and beyond, particularly in the context of the recently enacted Nature Restoration Regulation and in initiatives such as the EU Mission Restore our Ocean and Waters. Underpinned by this policy context, this research supports the creation of blueprints for replication and upscaling of innovative solutions to nature restoration.

## Introduction

The impact of the human population on the planet has resulted in multiple environmental and climate challenges that we need to tackle immediately (Fletcher et al., 2024). There is ample evidence that the planet's ability to meet society's ever-increasing demands is declining (Cradock-Henry et al., 2018; Chaplin-Kramer et al., 2019) and that large-scale transformational change within multiple sectors is urgently required to reverse this trend (Fischer and Riechers, 2019; Chan et al., 2020; van der Plank, 2024). Paramount in our response to these crises is the need to restore and protect biodiversity (Aronson and Alexander, 2013), not just for its intrinsic value but also to ensure it continues to sustain society through the provision of essential ecosystem services such as carbon storage, flood protection and food provision and through its role in economic activity (Turner et al., 2009). While the role of industry, governments and academia in this effort is often emphasised, the crucial part played by local stakeholders and

communities in biodiversity protection and restoration is by contrast usually underestimated (Davis and Slobodkin, 2004; Higgs, 2005). Many authors are now calling for 'transformational change' or 'transformational adaptation', and although the precise definitions of these terms remain somewhat vague (Vermeulen et al., 2018; van der Plank, 2024), it is clear that they must include a role for local communities and stakeholders (Holl, 2017). Crucially, this role should not be one of 'receiving' predetermined changes and adaptation approaches from the top down, as is often the case (van der Plank, 2024) but instead ought to be one of active participation in the process. Indeed, Aronson et al. (2020) suggest that the provision of 'training and capacity building opportunities for communities and practitioners' is one of six proposed actions to support the UN Decade on Ecosystem Restoration.

Working with stakeholders and communities at a local level would appear to be essential, but the issue of scale needs to be considered. Many biodiversity projects and scientific restoration studies work at smaller scales which are seen as being too discrete, difficult to replicate (Holl, 2017) and therefore not the answer to achieve the large-scale transformation that is needed. However, large-scale restoration can actually displace land users and jeopardise the provision of ecosystem services (Cardinale et al., 2012). To solve this conundrum, there is a need to view it through the lens of social ecological systems, which link the social and environmental aspects of our world in many complex ways (Fischer et al., 2020). Here, the inclusion of effective community-led action has the potential to overcome the issues of scale because achieving real transformational change requires not only replication of successful strategies, but also systemic change from a social, legal and regulatory perspective, among others (Chan et al., 2020).

When a 'top-down' approach to restoration is used, whereby governmental authorities or institutions make decisions about restoration goals, approaches and management and communities are simply the 'recipients' of restoration actions, there is a strong disconnect between communities and those in charge (e.g. Murcia et al., 2016). By contrast, employing a 'bottom-up' approach, whereby those living and depending on an area have a key and active role in the process (Holl, 2017), has the power to make robust connections and scale up efforts. This involves confronting the challenging work of listening to, understanding and integrating the perspectives of many different stakeholders whose lives will be affected by such measures. While this work may take longer and be considerably more complex and difficult to control, a review by Fox and Cundhill (2018) suggests that excluding a social dimension to nature restoration ultimately risks a lower success rate. There are already examples in the scientific literature of successful community-led projects in different ecosystems globally (e.g. Fraser et al., 2006; de Souza, 2016). That said, for communities at the start of the process, there are few accessible resources for them to leverage in trying to begin a biodiversity project at all. The success and continued legacy of community biodiversity projects depends largely on the sense of ownership and agency that communities have over their projects and initiatives and the sense of actions being rooted in what local populations need and envision (Cradock-Henry et al., 2018).

Nonetheless, the success of such projects also relies on the involvement of governments and statutory authorities so that multi-stakeholder collaborations are required. There is much evidence that participatory approaches to multi-stakeholder collaboration hold the best chances of success, mandating the involvement of stakeholders and enabling a joint development of approach from the beginning (van Drooge and Spaapen, 2017). In the field of sustainability science for example, participatory approaches to such

collaborations can generate a better problem definition, identify more suitable solutions, and enhance understanding and knowledge (Huttunen et al., 2022). However, what is potentially more powerful is the impact this participation has on connecting the everyday lives and actions of citizens, empowering them to engage with and act for sustainability (Fung, 2015; Huttunen et al., 2022). For example, it is acknowledged that community-based conservation (CBC) is a highly effective approach globally (Ellis & Mehrabi 2019), with long-standing calls for its use more widely (Chapin 2004) and frameworks developed to support its use (e.g. Mahajan et al., 2021). A comprehensive review by Fariss et al. (2022) showed that certain factors played a significant role in the success of CBC projects, such as ensuring they (1) took place in national contexts supportive of local governance, (2) confronted challenges to collective action, (3) encouraged diversification from an economic perspective, and (4) invested in capacity building. To facilitate a broader take-up of such bottom-up approaches among multi-stakeholder groups, there is a need to identify the most important elements of successful community-led collaborations from a practical perspective and to make this knowledge available in a truly accessible way to enable replication of success and a real and lasting impact on biodiversity.

This paper aims to characterise the successful multi-stakeholder collaboration that led to the development of a community-led nature reserve at Harper's Island Wetlands, Co. Cork, Ireland. The research was designed to align with a participatory evaluation research methodology, which actively involves stakeholders in the evaluation process (Cousins and Whitmore, 1998) aiming to jointly 'assess and monitor the success of a project or program intended to bring about positive changes in a community' (Park and Williams, 1999). The use of participatory evaluation facilitates mutual learning as well as supporting the relevance of the work and the approach taken to it. In transdisciplinary collaborations and public participation processes, participatory evaluation has the dual role of improving the collaborative understanding of the joint process as well as enhancing progress towards a common societal goal (Drooge & Spaapen 2022). This approach taps into, and identifies, the varying motivations and values that are at play for different people when they engage in nature conservation efforts (Chan et al., 2016), rather than assuming that everyone is coming from a nature-centric (intrinsic value) or human-centric (instrumental value) perspective. The chances of success of projects focussed on our current environmental crises can be enhanced by (among other things) increasing the intensity and extent of engagement efforts within those projects (Ferguson et al., 2022). This research was conducted such that members of the community in Harper's Island were active participants and collaborators in the work throughout, with outputs being co-designed and co-created with the community.

### *The A-A Agora project at Harper's Island Wetlands*

At present, there are a number of major initiatives being implemented by the European Union which focus on innovative approaches to nature restoration and climate resilience that emphasise the crucial role of community participation and multi-stakeholder approaches. The European Green Deal (EGD) for example, acknowledges that the protection and restoration of biodiversity and ecosystems is one of the key elements needed to foster transformative change (European Commission, 2019). This has been bolstered by the proposed Nature Restoration Law (European Commission, 2024). Crucially, the EGD also calls out that 'citizens are and should remain a driving force of the

transition to sustainability.' The EU Mission Restore our Ocean and Waters[1] has the aim to protect and restore the health of oceans and waters through research and innovation, citizen engagement and blue investments. In response to these initiatives, among others, the Horizon-Europe-funded project Atlantic-Arctic Agora (A-AAgora)[2] seeks to: (1) respond to the need to protect and restore marine and freshwater ecosystems and biodiversity, (2) protect valuable ecosystems located in coastal communities particularly vulnerable to climate change impacts, and (3) mitigate the effects of climate change, while promoting societal well-being. As part of this effort, the project seeks to identify examples of innovative approaches to nature restoration, to create blueprints and roadmaps that enable them to be replicated more widely and, in this way, facilitate a real and tangible contribution towards transformational change. The current research was funded by the A-A Agora project and aimed to trace and document a long collaboration between a community, a local authority and an NGO who, over 30 years, have undertaken a variety of nature protection and restoration activities at Harper's Island Wetlands.

### Study site: Harper's Island Wetlands

Harper's Island Wetlands[3] lie within the Cork Harbour Special Protection Area (single-page application [SPA] Site Code 004030) in Co. Cork, Ireland (see Figure 1). It is owned and managed by Cork County Council in partnership with BirdWatch Ireland,[4] the Glounthaune Community (Glounthaune Community Association/ TidyTowns[5] and Glounthane Men's Shed[6]), and the National Parks and Wildlife Service.[7] Harper's Island is a small (30 ha) low-lying island in the Glounthaune Estuary/Slatty Water complex, in the northern section of Cork Harbour. It is close to the residential area of Glounthane and is easily accessible from the road network as well as via a rail connection from Cork City. The low-lying northern section of the island is influenced by the surrounding tidal estuary through an old sluice point[8] (see Figure 1). Hence, the immediate vegetation is of brackish grasslands developing into successional salt marsh. The island was farmed for centuries. A reference to its use as arable pasture dates back to the 1600s (Éireann, 2025), with first edition 6-in. maps created in the early-to-mid 1800's recording agricultural-like boundaries and established transport connections to the mainland (Éireann, 2025). It has long been recognised as an extremely important safe feeding and roosting refuge for many species of wintering waterbirds. For example, at times during the spring months, peak black-tailed godwit counts can exceed 2,000 birds, which represents over 4% of the global population.[3] The broader area also supports populations of national importance of shelduck, teal, little grebe, golden plover, dunlin and redshank. Figure 1 shows the location of Harper's Island Wetlands in Co. Cork Ireland. It also provides a satellite image of the area, taken in 2003 before the work began on the site. Note that the farm infrastructure was no longer active at the time of the image, as

the site had come into public ownership in the late 1990s (and still belongs to Cork County Council) to facilitate the construction of a major regional road (N25) on the southern end. It was at that point that the site became 'unused' land, but continued to be of critical importance to many bird species.

The idea of creating a nature reserve at Harper's Island was first raised by the Cork branch of BirdWatch Ireland back in 1994 when they made a submission with the support of community members to the Cork County Council. In the intervening 30 years, a huge amount of work has been done by many to achieve what is today a thriving nature reserve open to the public, supporting large numbers of waterbirds and a wide variety of other plants and animals. The reserve is run by a multi-stakeholder steering committee composed of local community members (from Glounthane Community Association and Glounthane Men's Shed), representatives from the Cork branch of BirdWatch Ireland (an NGO), together with elected and staff members from Cork County Council.

This research employed a participatory evaluation research approach (Cousins and Whitmore, 1998) to engage with the Harper's Island multi-stakeholder group to (1) document the work undertaken by the stakeholder group and the impact it has had on Harper's Island Wetlands from the point of view of its land use and development and (2) identify the elements of success as well as the challenges encountered by this community, so that this case study can contribute learnings for wider replication and upscaling.

## Methods

### The participatory evaluation approach

This article draws on participatory evaluation research approaches (e.g. Cousins and Whitmore, 1998; Park and Williams, 1999) to a single case study of community-led restoration action, employing qualitative methods. Participatory evaluation research is 'an activity aimed at assessing and monitoring the success of a project or program intended to bring about positive changes in a community' (Greene, 2006). The study design, approach and outputs in this research were co-created with the steering committee of Harper's Island Wetlands and the co-creation process was facilitated through a series of engagement steps as shown in in Figure 2 and described in detail below. Through these engagement steps, data about the history of the reserve and how it developed was collected directly from the steering committee, from volunteers and from visitors. The aim was to have all those involved reflect upon the journey they had undertaken and to identify key learnings from that journey. This participatory evaluation research approach increased the likelihood that findings would more accurately reflect the lived experience of this community-led project. The qualitative data collected were then analysed through a reflexive thematic analysis process (Braun and Clarke, 2022), which allows a balance of flexibility and systematic rigour. This method also acknowledges that there is a relationship between the researchers and the participants and emphasises the deep interaction the researcher has with the data and their direct influence on the study. It recognises and values the researcher's subjectivity as the primary way to discern meaning from data.

### Engagement steps

An *initial meeting* took place in March 2023 between the two members of the research team and the community (four members) at Harper's Island Wetlands arranged and facilitated by Cork

---

[1]https://research-and-innovation.ec.europa.eu/funding/funding-opportunities/ funding-programmes-and-open-calls/horizon-europe/eu-missions-horizon-europe/ restore-our-ocean-and-waters_en (last accessed 3 April 2025).

[2]www.a-aagora.eu (last accessed 3 April 2025).

[3]https://birdwatchcork.com/about-harpers/ (last accessed 3 April 2025).

[4]www.birdwatchireland.ie (last accessed 3 April 2025).

[5]https://glounthaune.ie/ (last accessed 3 April 2025).

[6]https://menssheds.ie/sheds/glounthaune-mens-shed/ (last accessed 3 April 2025).

[7]https://www.npws.ie/ (last accessed 3 April 2025).

[8]https://birdwatchcork.com/wp-content/uploads/2020/02/2019-07-Conserva tion-Management-Plan-2020-2024.pdf (last accessed 3 April 2025).

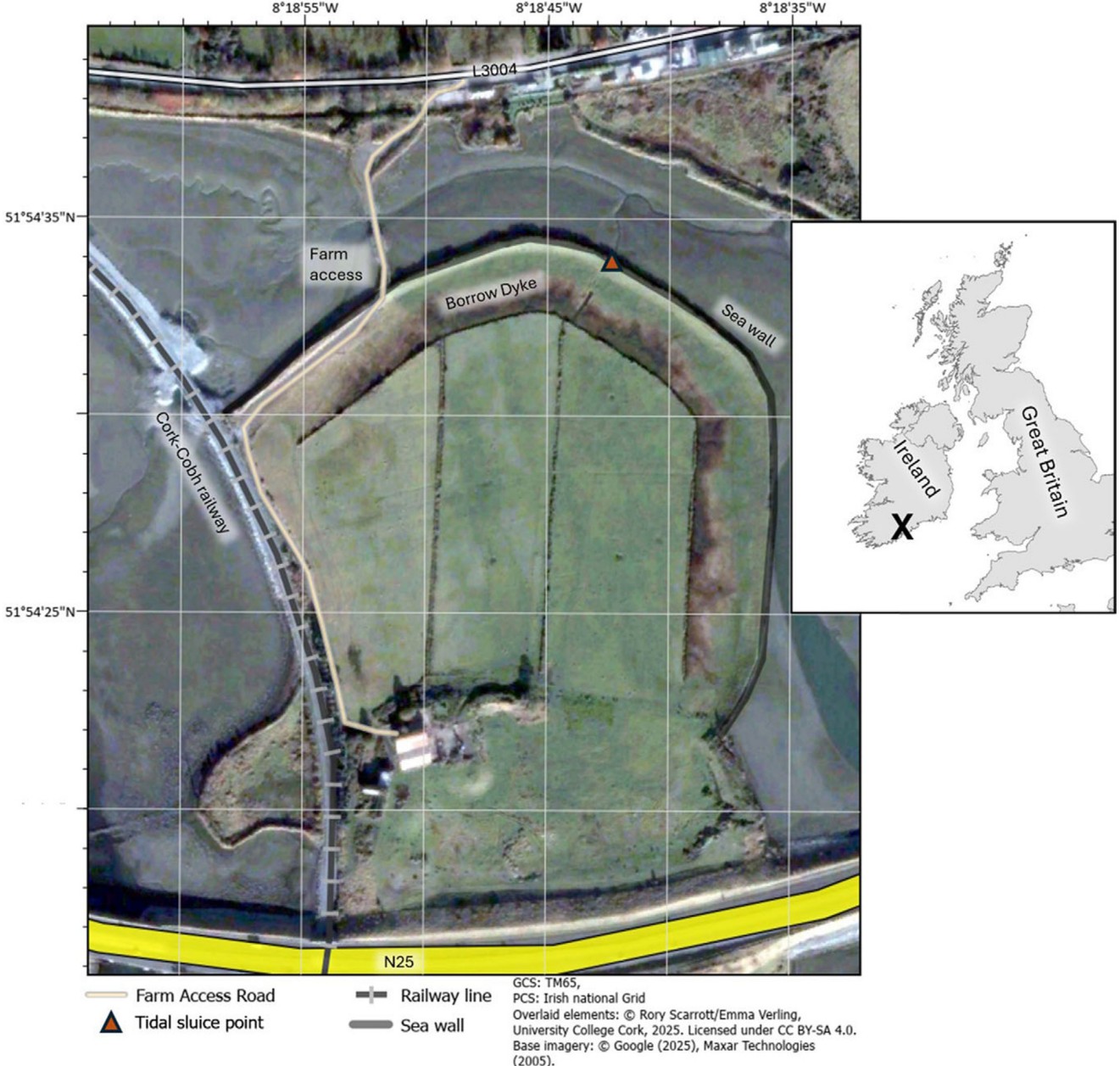

**Figure 1.** Location and pre-intervention image of Harper's Island in 2003, showing key transportation and water management infrastructure in place at the time. Underlying satellite imagery was acquired in November 2003 by Maxar Technologies (distributed through Google Earth) and has been geo-referenced. The location of Harpers Island is indicated by an X in the insert.

County Council (two members) during which the different groups were introduced to each other and the work they were doing. A *workshop* was then arranged by the research team and took place in August 2023 at the Glounthane community centre. There were 13 participants in total, with representatives from the Harper's Island steering committee and the research team. At this point, an external consultant with experience in engaging and working with communities was also engaged. The purpose of this workshop was for everyone to co-create a way of working together, to share ideas, build trust and brainstorm. Informal *walking interviews*[9] took place in September 2023 between the engagement consultant and

[9]https://sru.soc.surrey.ac.uk/SRU67 (last accessed 3 April 2025).

two key community members. This was an opportunity for the consultant to see and experience Harper's Island as well as to build trust. The next step in the process was the *semi-structured interviews*. A one-to-one online or in-person interview (as desired) took place between each steering committee member and the engagement consultant during October and November 2023. Eleven interviews were undertaken in total – five members of Birdwatch Ireland, three community members from Glounthane Community Association/Glounthane Men's Shed, one elected member and two members of staff from Cork County Council participated. All members were invited to participate in individual or group interviews and offered a choice of in-person or online interviews (between August and October 2023). Most were interviewed online, with some community members being interviewed in person and

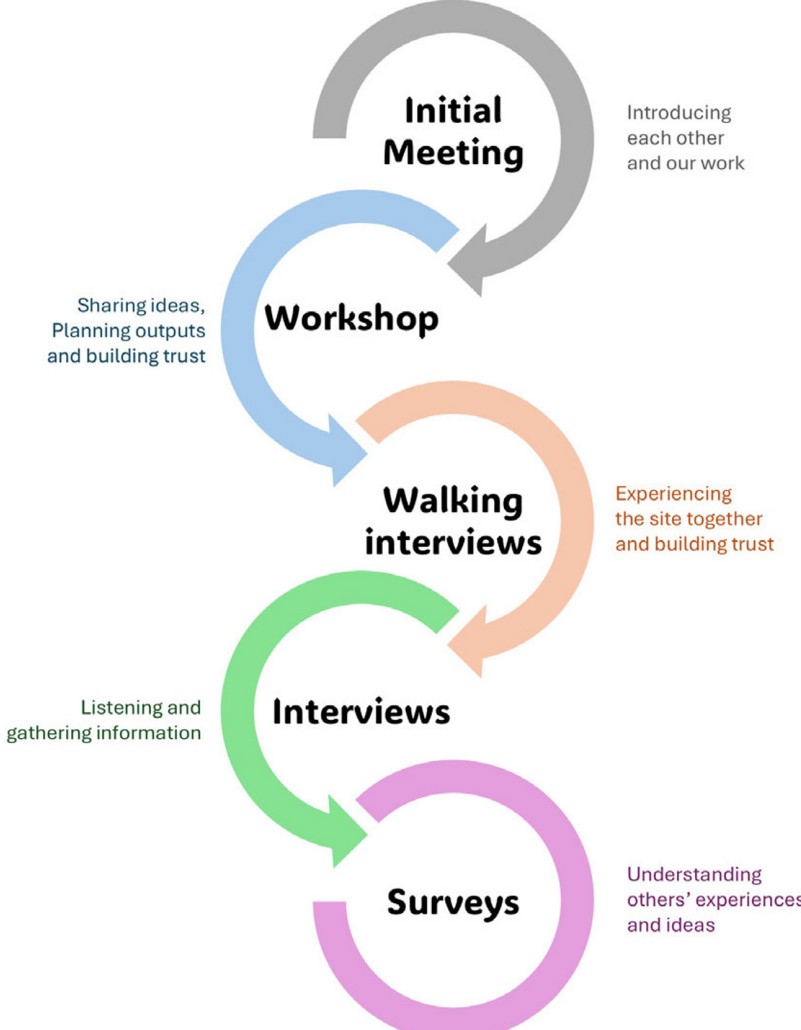

**Figure 2.** The engagement steps taken throughout this study to support the participatory evaluation approach with the Harper's Island stakeholders.

some choosing to be interviewed while walking through the site. The interview protocol focussed on understanding the steering committee member's history of involvement, experiences of the development of Harper's Island Wetlands, their thoughts on the challenges and successes encountered along the way as well as their hopes and ambitions for the future of the area. There was also some time given to their perception of the relationships between the different groups involved in the steering committee and how the evolution of these relationships had enabled the success of the project and their impressions of the impact the work has had, particularly within the broader local area of Glounthane and East Cork. The semi-structured nature of the interviews allowed for free-flowing conversation, created an open space for discussion and dialogue as well as leaving space for ideas and thoughts to emerge and to encourage honest reflection. This qualitative approach allowed complexities to be captured and provided deeper insights. The questions used as a basis for the semi-structured interviews were as follows:

1. Why did you have the idea to create Harper's Island, and why in this area? (origins)
2. How did you get involved? (motivation of individual gaining context)
3. Tell me a little bit about your skills, interests and involvement. (background and expertise/skill sets, available time, capacity)
4. How did the project develop, who was involved and when? (confirming activities in detail, timeline, group development and members of the group including gender/age)
5. How do you find working in partnership and collaborating with local residents, local authority and BirdWatch Ireland? (Benefits and challenges, understanding decision making, group processes, funding)
6. What were the challenges and lessons learned and how were they resolved?
7. Tell me about one stand-out moment for you.
8. Describe the process of communication with the wider community and other stakeholders. (awards and recognition)
9. What is your vision for Harpers Wetlands (from all their perspectives) and link to the community? (reflection and vision time)

Finally, *surveys* of volunteers (online) and visitors (in person) were conducted. The purpose of these was to gain an insight into the experiences of the wider public using the reserve. A total of 20 visitors to Harper's Island were surveyed in person on 2 December 2023 using a set of pre-defined questions exploring the reasons for

their visit, their impressions of the site and any suggestions they might have for the site. These questions are provided in the supplementary information. In September 2023, 14 volunteers at Harper's Island answered a set of pre-defined questions via an online survey, which investigated their motivations for volunteering and the benefits they derived from this work. These questions are provided in the supplementary information.

The data gathered from the semi-structured interviews, both online and in person, were voice recorded. These recordings were subsequently transcribed, and thematic analysis was applied to identify key themes. Surveys with visitors and volunteers were analysed to provide additional feedback and insights into the case study. The results presented here were summarised into an 8-page reflective brief for the community and for wider dissemination, in particular among other communities and local authorities. Satellite imagery provided an additional source of data in relation to changes in land use over time. Drawing on mixed methods approaches (interviews, survey data and satellite maps in this case) can help to triangulate evidence and demonstrate rigour in case study evaluation.

## Findings

### Development of the Harper's Island Wetlands

The semi-structured interviews undertaken during this research highlighted that since Harper's Island was identified as an important birdwatching site in the 1990s, a huge amount of work has been invested by several different stakeholders and stakeholder groups to create what is now a thriving, accessible and growing nature reserve, which is open to the public. Figure 3 provides an image of the site taken in 2024, indicating the interventions and illustrating the development of the site since the work began, while Table 1 provides a list and timeline of those interventions. When contrasted with the image of the site taken in 2003 (see Figure1), it can be seen that the area has been transformed considerably, both from an environmental and ecological perspective but also in terms of accessibility and outreach. The interventions undertaken range from restoration and conservation activities, as well as educational resources and infrastructure to facilitate public access (see Figure 4 for some examples). A network of nature trails (visible in Figure 3) and two bird hides have been constructed, as well a large amount of educational bilingual signage (in English and Irish) throughout the hides and along the trails. Educational materials have also been produced for use with school groups. A series of restoration and conservation activities were undertaken to enhance the site. These include the construction of 'scrapes' (shallow ponds of less than 1 m depth) to facilitate water birds, installation of nesting sites for Sand Martins, the creation of a reed bed, ponding and pollination areas, bee scrapes, as well as the implementation of regular bird counts (which are available on the open-access eBird platform) and the introduction of grazers to control invasive species and maintain the wetland for wintering birds. These interventions were all valued by visitors, who (during the survey) cited the sense of peace, ease of viewing birdlife and accessibility as motivators for their visit, with most of them hearing about the reserve through word of mouth.

### Elements of success

Thematic analysis of the semi-structured interviews and surveys showed that the success of this bottom-up wetlands project was a result of 10 key ingredients and processes, all of which were considered to contribute to the overall 'power of partnership'.

These are detailed below and are also summarised in a user-friendly document co-created by the Harper's Island steering committee and the authors of this article.[10]

1.  *Collaboration.* A project steering committee was established and this operated in a very collaborative manner between the mix of partners. Because of this active and practical collaboration through which results were achieved collectively, partners developed a mutual respect for each other's skills and offerings. This had the additional benefit of impacting and broadening individual members' motivations. For example, considerable cross-learning about the environment and biodiversity – beyond the scientists and birders – brought about wider engagement with issues such as climate change and fostered an appreciation of the wetlands and the bird life there. This wider appreciation then further reinforced the passion and commitment of the committee. Partners all strongly argued for their viewpoint but were not territorial and all were seen as being politically astute. This facilitated working with key institutions and departments, building good relationships with local councillors, senior Cork County Council staff, local industry and regularly bringing visitors to the Wetlands. All of this increased the profile and positive image of the project, making it easier to access funds.

2.  *Mixed skill sets.* The Harper's Island steering committee was composed of a wide variety of different groups and people, all of whom brought unique skillsets and expertise to the table. As a result, it was regarded as a high-functioning working group. Ultimately, the mixed-skills element brought with it a balance of support, knowledge, practical skills (e.g. construction skills, farming knowledge), relevant expertise (e.g. ecological knowledge from BirdWatch Ireland members) and local authority powers (e.g. access rights, ownership) to undertake appropriate biodiversity protection and restoration while also allowing access to the public, in harmony with nature. However, interviewees did acknowledge that project partners should be chosen with care as achieving the right mix of partners was considered to be absolutely critical to success.

3.  *Passion and commitment.* The passion and commitment to the project were a common denominator among all partners, and all interviewees cited these as core elements of success. While each partner's commitment was motivated for slightly different reasons (some were focussed on community, others on nature and birds and others on providing amenities), all of these motivations were complementary and ultimately had an additive effect. Because the buy-in, passion and commitment from all members was broad and significant, the workload was consequently shared among all.

4.  *Determination and resilience.* The Harper's Island project has been ongoing for a long time (over 20 years). This research identified that the determination and resilience of the various partners contributed strongly to its success. Despite challenges along the way (which are detailed in the next section), there was a 'win' or a benefit in this project for all partners. This tapped into partner's motivations and contributed to developing the project successfully. For example, among the wins achieved were that the community got a fantastic amenity and national recognition for their work, BirdWatch Ireland met their goals around conservation and increasing the availability of bird habitats. Meanwhile, Cork County

---

[10]https://www.marei.ie/project/atlantic-arctic-agora/ (last accessed 3 April 2025).

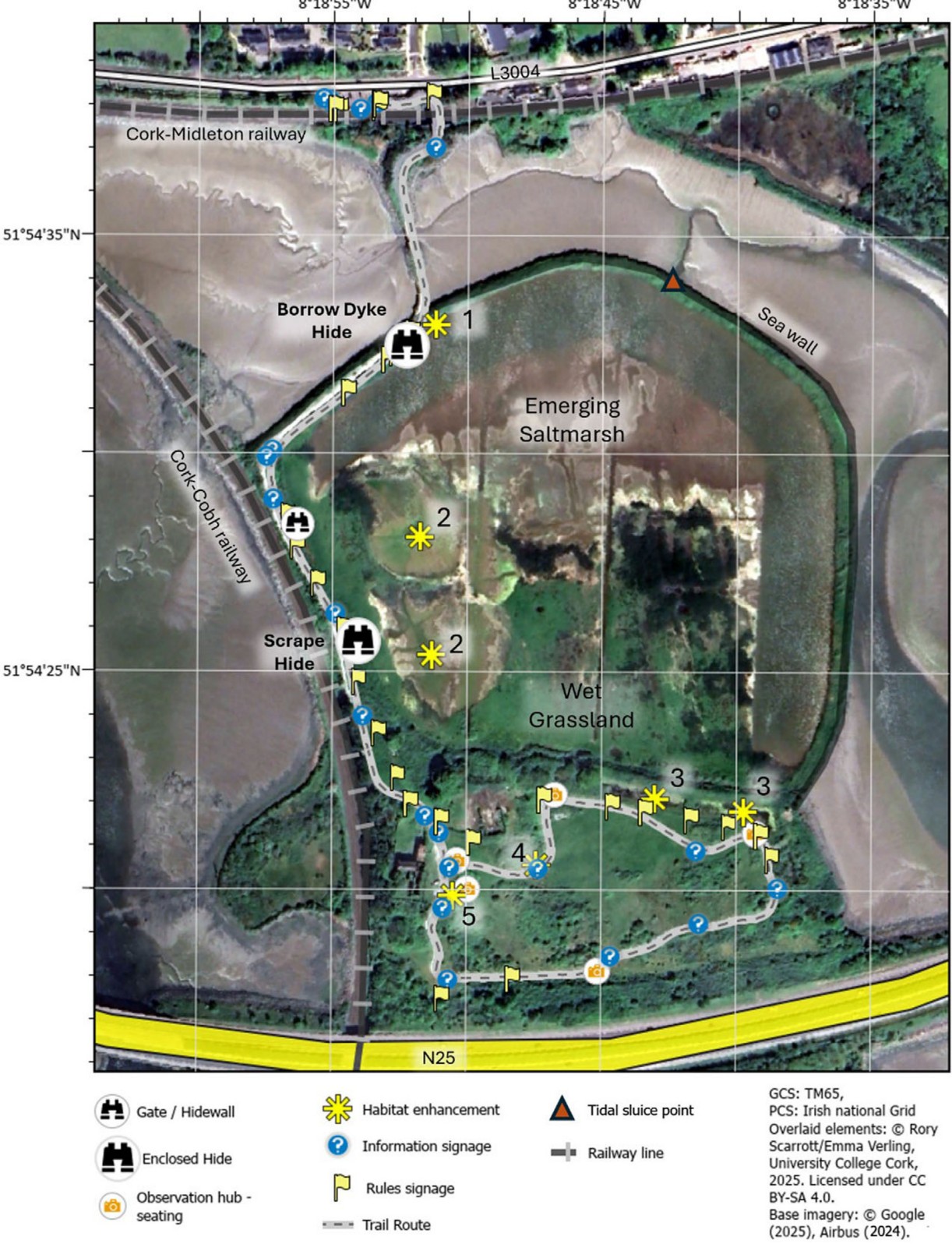

**Figure 3.** Interventions implemented and installed by the steering committee and volunteers at Harper's Island to complement emergence of wetland habitats within the bounds of the sea wall. Interventions were mapped in February 2025, and include trails, hides, signages, benches for observation, and habitat enhancements. Habitat enhancements are (1) two sand martin nesting banks, (2) two wetland scrapes, (3) two marsh ponds, (4) mining bee scrape, and (5) a dual biodiversity enhancement/education woodpile for invertebrates. The map does not include pollinator mounds which have also been installed. Underlying satellite imagery was acquired in May 2024 by Airbus (distributed through Google Earth) and has been geo-referenced.

**Table 1.** A timeline of activities that have taken place at Harper's Island Wetlands since the early 1990s

| Year | Activity/action |
| --- | --- |
| 1993 | Site farmed until the 1990s. Research study highlights the importance of the site as a birdwatching site. |
| 1995 | Bird counting commences at Harper's Island. |
| 2001 | Preliminary discussions take place on the development of a nature reserve at Harper's Island. |
| 2009/10 | A chance meeting between local residents and bird watch experts means meetings are initiated again to develop a reserve. Horses' grazing assisted the management of the wetlands. BirdWatch Ireland and Glounthane Community Council present a proposal to Cork County Council. |
| 2014–2017 | Environmental Impact Assessments were conducted, and the first birdwatching hide and first scrape were built. |
| 2017 | A steering committee was formed for the first time. Official opening of the site occurred in December 2017. |
| 2018 | A 5-year conservation management plan was put in place. A second scrape was built, attracting over 20 bird species. Road signage was installed for the site. |
| 2019/2020 | A nature trail and reed bed were created. Education and information boards were installed in the hides. A feasibility study into the creation of a visitor centre at the site was completed. A formal agreement was reached around horse grazing at the site. The second bird hide was built by the Glounthane Men's Shed. |
| 2021 | The first Sand Martin bank was constructed. |
| 2022 | Improved groundworks were undertaken as well as a pedestrian crossing to the site, car parking spaces and a bike stand. An education pack for schools was created and signage installed on the site. |
| 2023 | Construction of a second Sand Martin bank began. Students from University College Cork and local schools began projects on the site. A study of the old farm building on site began to establish what sort of development could be possible in the future. |

Council got to develop a nature reserve on their land. To achieve all of this, all partners remained focussed and committed for the long term.

5. *Role of facilitator.* It was acknowledged by interviewees that the role of Cork County Council, who acted as an anchor and adjudicator on the steering committee and invested funding, was crucial in stabilising the project and in aligning different partners' goals and visions. This stabilising effect kept the project focussed and allowed issues to be resolved in a systematic manner as they arose. The facilitator also managed expectations in terms of processes and limitations and used their position as the land owner to great effect. Everything carried out on the site was subject to planning and sustainability principles and the community benefited from becoming familiar with using this measured approach, which also built capacity and knowledge within the community.

6. *Taking advantage of opportunities.* One element mentioned by several interviewees was the willingness to identify and seize opportunities when they arose. In that sense, the committee had to be flexible and dynamic. For example, having the readiness, practical skills and willingness to build the bird hides as soon as assessments were completed, and approval given was key to keeping the work on track. The re-opening of

a railway line from Cork to Midleton in 2009 facilitated the construction of a bridge to Harper's Island, establishing an easy access point to the area. Furthermore, having access to the local authority and expert knowledge within it in terms of planning and habitat legislation was something the group was able to capitalise on. All interviewees identified 'serendipity' as playing a role – even during the COVID19 pandemic, for example, local residents had a greater opportunity to enjoy and learn about the special amenity they had on their doorstep.

7. *Contracting local groups.* When undertaking the practical work on the site, the approach taken by the Harper's Island steering committee was to contract local groups and individuals to undertake the work. For example, Cork County Council contracted the Men's Shed and Cork Branch of BirdWatch Ireland to ensure the practical work involved in building the bird hides was completed. This represented very good value for money, but more importantly, it deepened relationships, trust and local ownership of the project.

8. *Doing practical work together.* A considerable amount of practical work was undertaken in the field by the community and steering committee together. Undertaking this practical work – for example, the construction of bird hides – was considered very effective in building trust and confidence in the partnership, facilitated members to resolve challenges and differences as they arose and to work in harmony in a functional way.

9. *Taking the time to demonstrate what works.* Work was undertaken within the reserve while maintaining the site open to the public. Because of this, work was done carefully and mindfully, progressing the project in phases and developing a well-informed conservation management plan. This slow and steady development and unfolding of the nature reserve allowed time to demonstrate how different approaches were working and where they could be changed or improved.

10. *Rewarding success.* Interviewees felt that acknowledging successes created further success at various stages and also motivated and energised partners. Making the effort to apply for and subsequently winning awards acknowledged the success of this project and gave it legitimacy, publicity and further buy-in.

## Challenges and barriers to progress

The interviews and surveys identified some considerable barriers and difficulties encountered during this project. It was considered very important to acknowledge these and to develop mitigating measures to deal with them.

1. *High time commitment.* The Harpers Island Wetlands project was demanding of significant time for both volunteers and staff. The availability of local authority staff was limited, and in reality, there will always be many community projects seeking support from these agencies. Therefore, in their forward planning, communities should not underestimate the time commitment required to realising these types of projects.

2. *Lack of volunteers.* All partners stated that they would like to have more volunteers involved in the Harper's Island Wetlands project, especially young people and more diverse groups representative of broader society. However, the effort involved in increasing and engaging volunteers is considerable and it very likely requires a structured plan of work. This is something that the community would like to improve and focus on in the future.

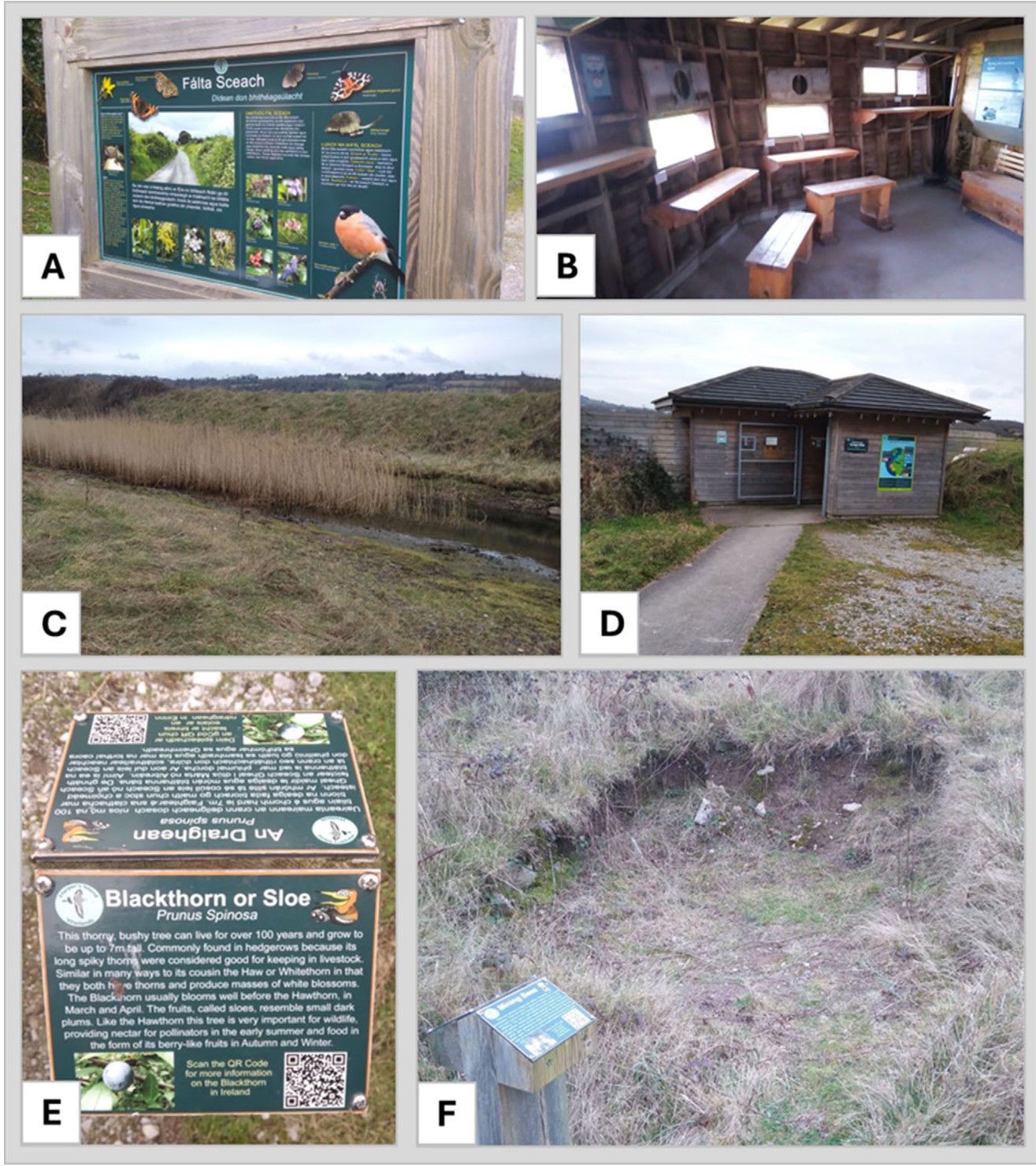

**Figure 4.** Examples of interventions surveyed on 10 February 10 2025. Shown are (A) large dual language information on habitats; (B) inside the Borrow Dyke Hide showing the observation area, and installed information content; (C) one of the developing marshland ponds with developing reed habitat; (D) the Scrape Hide with installed information content for visitors along with a sample of the durable trail installed; (E) smaller dual-language information signage highlighting key aspects of the native ecology; (F) a habitat enhancement in the form of a simple scrape to expose a soil face for mining bees to use, coupled with a small information post about them.

3.  *Need for long-term strategy* The absence at present of an overall long-term development strategy for Harper's Island Wetlands was seen as being a barrier to progress, and it is something the community would like to work towards in the future.

4.  *Rising age profile.* The age profile of the steering committee at Harper's Island Wetlands is rising and there was a feeling that a legacy plan needed to be put in place in order to secure the future of the site. Many of the partners were acutely aware of this issue and agreed it needed further discussion.

5.  *Dealing with power dynamics.* There was an honest acknowledgement that the shifting power dynamic over time within a large group can cause tensions and conflict, which in some cases can cause the collapse of a project or group. This reality was acknowledged and dealt with in an open and solution-focussed manner and is part of the journey of a community-based project. Being considerate of each other's skill sets and being open to professional advice was an area of strength of the steering committee, which then enabled easier and more appropriate decision making.

## Discussion

In examining the learnings that emerged directly from this community, a number of insights emerge. The development of the nature reserve at Harper's Island highlighted the real impact of fostering a sense of ownership and connection to the environment through restoration practices. Actions like these that change mindsets constitute a 'deep leverage point' (sensu Fischer and Riechers, 2019), which have the potential to effect the transformative change needed to achieve a socially just and ecologically sustainable future. One participant stated:

> Summer 2017 and almost immediately we saw it being used by lots of birds, so it was working, because I suppose, yeah, that was the first kind of work we did and we didn't know exactly what we were doing, and whether would it work, and to see it working straight away and was such a success, it was a big standout moment for me, yes. (Participant 5)

> .....building that Sand Martin Bank then the first summer, 24 Sand Martin families occupied the 24 spaces, and people coming down and looking at them and seeing them … that was a YES (moment)! (Participant 1)

Local communities often have the potential to mitigate climate impacts and reduce disaster impacts, both from a human and financial perspective (Dale et al., 2020) and the work at Harpers Island exemplifies this. Despite continual development of the surrounding area from a socio-economic perspective (e.g. the addition of a new railway line and housing visible in Figure 3 compared to Figure 1), the site itself has become an amenity for the public as well as an area where biodiversity protection is prioritised. The study highlights, however, that the creation of such a multi-stakeholder group does not necessarily have to begin with a global issue such as climate change or biodiversity loss; these were not catalysts for setting up the reserve at Harper's Island, yet the interventions at the reserve directly tackle these issues. All partners were united in their commitment to the site and its development as an amenity for the community as a whole. In fact, the robust collaboration between partners, and the interplay between their different motivations, created a positive feedback loop which continuously strengthened the partnership. This same finding has been reflected in studies of community-led actions focused on the Sustainable Development Goals (SDGs)[11] which together have been found to have addressed all SDGs at a local scale (Henfrey et al., 2022). This study therefore underlines the need to consider and promote local focus and individual motivations rather than attempting to impose global drivers such as the climate or biodiversity crises. The visitor's survey at Harper's Island showed the impact and power of local pride, knowledge and promotion, because most visitors had heard about the site through word of mouth and not via other means.

The impact of having a practical and functional relationship between communities and local authorities/government agencies is also something that emerged strongly from this work. The model of a steering committee 'anchored' by the local authority was found to be very effective and stabilising for the project. One of the participant comments illustrated this:

> The big success in this programme I think is the Council, that they actually facilitated the group and helped manage it. They don't have a really strong agenda other than to deliver the project as a whole. I'd say that 90% of the initiatives have actually come from the Community Association or from the Men's Shed, or from BirdWatch Ireland. The Council initiatives really have been in terms of how to get funding, and putting it onto the agenda, and getting the project moving, and then keeping it within the bounds of the regulations. (Participant 4)

In a wide-ranging review of biodiversity conservation measures, Cook (2024) showed that this finding is widely reflected elsewhere. Other studies identified that having the right governance and management structures are instrumental in achieving biodiversity protection (Dentoni et al., 2018; Boyle et al., 2021; Ward et al., 2024). Local authorities and government agencies should therefore strive to support communities as early as is possible, both through practical intervention and through support to identify additional funding. Although these relationships are not always easy to establish and maintain, and challenges were encountered in this case, this work also illustrated that mutually respectful collaboration is paramount. Multi-stakeholder collaboration in general requires dedication and resilience from all partners to achieve and maintain success (Mattessich and Monsey, 1992), and this has been shown to be a critical point for sustainability collaborations (Boyle et al., 2021). This research showed that support in the form of funding and dedicated staff time from within local authorities was also an effective enabler. For example, central funding could be more directed towards community officers, who actively engage in project management and funding applications. However, success could also lead to more demand for replication of activities and with limited human and financial resources. Therefore, while it is vitally important to manage expectations. Particularly for local authorities, a positive relationship between top-down governance and bottom-up action supported by policy can facilitate success and replication. Positive relationships between stakeholders emerged clearly from this work, as shown in the following quote:

> I think that … developing the trail and the second hide … was fantastic. You know that interaction with the lads on a weekly basis, going down, having the chats, figuring out what we were doing next, what was needed to be done this week, all of that work, and then getting the hide and the trail opened, you know, I have to say that was probably the highlight, yeah. I really enjoyed it, it was great. (Participant 2)

In this single case study, the creation of a user-friendly document (the approach favoured by the steering committee) to be disseminated to a very wide audience was the first step. The ultimate aim of documenting this case study was to facilitate 'scaling' whereby sustainability initiatives apply amplification processes to foster transformative change - amplifying within, out, and beyond (Lam et al., 2020). As well as scaling up and scaling out, increasing impact can also be achieved by changing values and mindsets, referred to as 'scaling deep'. Scaling deep can be an important conceptual tool for regional sustainability transitions (Boyle et al., 2024) where operationalising community empowerment has the potential to engender much more impactful climate action due to the adherence to local dynamics and enabled characteristics while still remaining connected

---

[11] https://sdgs.un.org/goals (last accessed 3 April 2025).

to national and European agendas. To facilitate scaling and amplification, we have identified four essential components that groups should consider when trying to establish a bottom-up project:

1. Assembling a core committee (which could be seen as a Community of Practice (Snyder and Briggs, 2003). The membership of this group needs to be considered carefully and should contain a mix of diverse skillsets appropriate to the task. Diversity of opinion and motivation within the group should be encouraged, but the key is that everyone in the group has a shared goal or aim, even if motivated to achieve this for different reasons. Such multi-stakeholder partnerships are already an area of increasing interest among those tackling sustainability challenges (Dentoni et al., 2018) and are likely to continue to be explored further in the coming years.

2. Establishing clear roles within the committee. The role of an anchor or 'adjudicator' was considered particularly important (in this case, that role was undertaken by staff from the local authority), as was establishing a culture of mutual respect, listening and open-mindedness. This could be seen as part of a collaborative governance approach, real-world examples of which have received a limited research focus, but which are receiving an increased interest and have the potential to be very powerful (Margerum et al., 2016), particularly in the face of sustainability and environmental challenges.

3. Creating a strategy for the project that covers short-, medium- and long-term timescales and associated goals. This strategy should include a risk register (identifying potential risks and associated mitigation measures), identify initial practical tasks with nominated people to lead them, consider communication with the wider community, identify funding sources or awards to apply for and include provisions for monitoring and updating the strategy to keep it current.

4. Beginning practical tasks as soon as possible, organising collaborative practical work on the ground (enabling groups to bond and socialise). Local contractors and groups (e.g. in this case the Glounthane Men's Shed) should be engaged where possible. Groups should be prepared to allow time for the work to develop and to consider pathways to assess success and impact of their partnership.

The above actions can be taken by communities or stakeholder groups in any jurisdiction or environment. There remains a major challenge to enable the learnings highlighted in this work to be communicated and emulated elsewhere. This is a crucial aim of projects funded within the EU Mission Restore our Waters and Ocean such as A-A Agora, CLIMAREST[12] and others. Over the coming years, a concerted effort should be made to create blueprints to upscale and replicate innovative approaches to nature restoration such as the one presented here. A strong policy focus on bottom-up, multi-stakeholder approaches is crucial. These need to be implemented at regional, national and European levels to achieve the goals of National, European and Global initiatives within the timescales required to tackle the many environmental and climate challenges facing hte planet.

**Open peer review.** To view the open peer review materials for this article, please visit http://doi.org/10.1017/cft.2025.10002.

**Acknowledgements.** The authors are very grateful to all the members of the Harper's Island steering committee and the wider community for their enthusiastic participation in this work. We would also like to thank Catriona Reid and Cathal Gannon for their assistance with visitor surveys. Finally, the manuscript was significantly improved thanks to the comments and observations of two anonymous referees, and we would like to thank them most sincerely.

**Author contribution.** E.V. – conceptualisation, writing: original draft preparation, writing: review and editing, visualisation, supervision, funding acquisition; M.P. – investigation, methodology, writing: review and editing; M.B. – writing: review and editing, conceptualisation, visualisation; A.D. – methodology, writing: review and editing; R.S. – visualisation, writing: review and editing, conceptualisation; D.Ó.S. – project administration, supervision, funding acquisition, conceptualisation, writing: review and editing; L.W.P. – project administration, writing: review and editing.

**Financial support.** This work was part of the A-A Agora project. This Project has received funding from the European Union's Horizon Europe research and innovation programme under Grant Agreement 101093956. Views and opinions expressed are those of the author(s) only and do not necessarily reflect those of the European Union or the granting authority. Neither the European Union nor the granting authority can be held responsible for them. This work was greatly enhanced by many interactions and discussions with the wider A-A Agora project consortium, and we would like to extend our heartfelt appreciation to them for this.

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
