## [Reviewer Report]

This is an interesting account of a community-based wetland restoration project in Ireland. The team has undertaken interviews with the steering committee, and surveys with volunteers as well as visitors. From this work they have resolved some key takeaways about the success of the undertaking, as well as cautions, and some practical advice. The project sounds wonderful. There may be some utility in the findings, if better contextualized, but this offering is not academically rigorous and reads superficially, thus is not suited for a ‘research article’.

I wonder if the authors might re-engineer it as a case study. In the instructions for authors a case study “should contain critical evaluation of the projects, well-developed discussions that provide constructive information, and recommendations for improvement within the field”. I see this as similar to what Society and Natural Resources calls ‘practice-based research’ papers, and that venue may be another good outlet for a paper with the goals and author list of this one.

Even if the authors re-engineer for a different paper type, they need to provide much more detail about the methods, consistent with norms in the social science fields they draw upon for their methods (e.g. survey effort, response rate, completion time, context, personnel, question design and justification, analytical process), report on those data in a more rigorous way (again, consistent with SS norms so that we can see from where their insight derives, reporting qualitative and quantitative), and engage with the substantial relevant literature in the field of environmental sociology and conservation social science in the front end and discussion. Claims are made that we have not been ‘shown’, e.g. that it “changes mind-sets”. There is no discussion or sophisticated interpretation, and it proceeds without framing beyond big picture ecology/SES ideas.

Overall, the communication is generally strong, with minor punctuation and other issues that will be caught by typesetters. The graphics require significant work to help us picture the wetlands in better detail (e.g. the size, the sluice, nearby land uses, etc.). The single data graphic is entirely inadequate and of unclear value, especially given the small sample size (and perhaps survey effort).

The work needs to rise above the case to create transferability, in part by being more transparent and connecting it to ongoing conversations in the field. This would do justice to what seems to have been an important exemplar of community-led conservation.

---

## [Reviewer Report]

Brief summary

The paper entitled ‘The power of learning from the bottom up: A blueprint for community-led biodiversity protection and restoration’ describes the successful implementation of a bottom-up developed coastal wetland restoration project in Cork County, southern Ireland. The paper specifically addresses the yet understudied example of a successful community-led biodiversity project, ‘in which communities are the architects of the action – as opposed to the recipients of it’. In their study, the authors use semi-structured interviews, online, and in-person surveys with different stakeholders from the Harper’s Island Wetland Project to derive ten key determinants for restoration success. The authors anticipate that these points may help upscaling bottom-up restoration projects, which are much needed in times of global climate change and increasing coastal pressures.

General recommendation

The paper addresses a relevant topic and makes a good start in identifying the need for more research related to community-led nature restoration projects, particularly with respect to success factors and potential barriers. The research question and the insights derived using the semi-structured interviews are interesting, relevant and certainly of interest to the wider readership of Cambridge Prisms: Coastal Futures. However, in my opinion, the paper is lacking scientific rigor. The manuscript does not adhere to the standard structure of a scientific paper, by missing out on crucial sections, such as methods, discussion and conclusions. Furthermore, no information about the details of the semi-structured interviews are provided, making it hard to replicate the findings. Furthermore, it remains unclear whether the results of the presented work are actually transferable to other regions, and how the findings may compare to previous studies. In my opinion, a discussion around these points is crucial for the paper to make an impact beyond the case study it presents.

In summary, while the research question and results are relevant and interesting, I believe that major revisions are necessary before the article can be published in Cambridge Prisms: Coastal Futures. Please find my detailed comments below.

Major comments

1. L101-105: How many more or less realized projects would we get by applying a community-led bottom-up approach? Because often, societal resistance is hampering biodiversity projects, which not necessarily points towards increasing project implementations using a bottom-up approach. Can you elaborate a bit more here? This point should also be picked up in the discussion section, as I believe it is very interesting and relevant within the context of your work.

2. L114: The bottom-up approach of the Harper Island Wetland project needs to be more clearly communicated in section ‘Background’. In my opinion, too little is known about the precise approach of Harper Island and how it differs from more classical top-down projects.

3. L177: The article is missing a methods section and too little information about the semi-structured interviews, as well as the online and in-person surveys is provided. The authors should justify the methods applied using exiting literature and then provide more details about the structure and questions of their interviews to enable the reader to better evaluate the outcomes of their work.

4. L377: The paper is missing a discussion section. In this section, the presented results study should be brought into the wider context of the existing scientific literature, which is not yet sufficiently well represented in the text. In my opinion, the authors should elaborate in the discussion...

... whether or not, and how, their results may be transferable to other regions/countries;

... how their results compare to the lessons learned from previous projects, which eventually helps finding out whether their points are transferable or not;

... how the well-known problem of societal resistance against coastal restoration projects, such as managed realignment, can be overcome using their ten key points.

5. Figure 2: The figure should be improved in terms of readability. Furthermore, I believe this figure and the text around it should rather be part of the results section, just like the point above stating that climate change was actually not a catalyst for setting up the restoration project, which I find very interesting.

Minor comments

6. Figure 1: A more detailed map of the area under investigation would be helpful for the reader. For example, an aerial photography/satellite image from the wetlands and the island.

7. L246: Just a suggestion: Why not make this point the first in the list, as the committee really was at the start of the project, right?

8. L18: ‘some of the challenges’: I think there is a ‘the’ missing, but I am not a native speaker, so pls check.

9. L392: ‘There is ‘a’ therefore a need to consider’: I believe there is an ‘a’ too much in there, please check.

---

## [Editor Report]

Dear Emma,

I have now received two reviews of your paper ‘The power of learning from the bottom up’. As you will see, both reports are very thorough. While they are broadly supportive of what you have been doing, both reviewers raise significant concerns over the presentation of the research. Both refer to a lack of rigour. There is a need to give more information on the Methods used and to better describe the field setting. There needs to be better engagement with the relevant environmental sociology and conservation social science literature at the front end and in a proper Discussion section. I am afraid that there is a great deal of work to do here in what would be a major revision of the submission. But this is an area where ‘Coastal futures’ is very keen to see papers in the journal and the full reviews give plenty of pointers as to how the manuscript needs to be re-written. I encourage you to ‘go again’ and submit a substantially revised paper.

Yours sincerely,

Professor T Spencer

Handling Editor

Special Issue: Transdisciplinary Science for Near-future Habitable Coasts

---

## [Reviewer Report]

The authors have done a solid revision here and provided me a lot more to chew on.

I’ve got a few additional comments inspired to strengthen it further. The key thing I think I’m lacking right now is a sense any existing literature on the successful elements of community-led conservation work. The authors draw up front on work about the importance of community leadership (not only engagement), but not on the factors that lead to success in such collaborations. There is some at the back end, but not up front. What do we already know going into this kind of work?

I’d revise the abstract to include the key insights in the discussion in lieu of the lists of results, which are not very novel. I’d remove “gain speed and momentum” too. There is an extra “the” in line 19 (using the authors’ numbering).

72-73 awkward sentence: precise definitions are nuanced? I’d say definitions are vague, or something similar.

159 I’d add a colon after the “to” before 1 and pull the “to” out of item 2 and 3 to clarify the reading.

Study site: I’d love to know a bit more about how long the site was farmed, what was farmed, etc. How did the government land purchase proceed? Understanding the historical context helps to know the degree to which such trajectories are affected by resistance to change for farming or cultural reasons. We see such resistance a lot where we are.

Figure 1: I used Google Earth to figure out where it was and where people lived relative to it, how they access it, etc. It feels like Figure 1 could include a scale between the two current maps (like a bit more of a zoom into the old Fig 1) to give us a better sense of the (human) context. One additional recommendation is to label on the map wherever possible to reduce the number of things in the legend (especially for folks like me viewing in black and white). For instance, label the national road on the map, since there is only one, and the local road, etc. This map won’t be used for navigation so coordinates aren’t important. Simplify as much as possible so the main themes are clear. These are nice maps, and the satellite image pre and post are really useful. But especially when there is a satellite underlay, all other opportunities for simplification need to be found. Where is the sluice gate?

Engagement steps: Maybe use initials to indicate who from the author team was involved in specific steps. How was authorship decided, given the community partnership?

Interview questions must be indicative. Surely not all were in at the ground floor, as indicated by 1. But maybe so. Also, many of them are not questions and should not have question marks.

289: Did you ask about disbenefits or challenges, too? Or only positive things?

317 add “of” before scrapes. Scrapes are a new idea for me. Thanks for the illustration for clarity.

320 How many horses were introduced? Where do they graze, how often, and where do they stay. Is the grazing controlled at all to avoid preferential grazing (unless they prefer invasives?).

Table 1 leaves quite a few things out that are included in the narrative, for instance the land purchase, etc. 2001 who was involved in the discussions. 2009/10 “birdwatching expert”. 2023 what is a conservation study of a building? Is it historic? Finally, what do the colours/shades mean?

Results would be enriched by some quotes. Bit anemic without hearing some of the voices in this qualitative study.

Lack of volunteers: Is the site on a public transit line? Or do people need a car to visit? Could this affect demographics?

490 I didn’t hear much about mind-sets being changed. Could a bit more evidence be given for this in the results? Seemed more as if like-minded people achieved something great.

Land acquisition is a big issue in my jurisdiction and I wonder if the authors would have any insight on ways to overcome barriers to public land purchase? And has the new landowner committed to maintain that sea wall? Or will it be allowed to degrade?

507 “addressed”

514 “relationship”

535-536 Any advice from the authors about the balance of ‘anchors’ and the benefits that come with them, and the community members?

574 How does the “risk register” emerge from data? And what is a risk register?

Some author names are misspelled (e.g. Risher instead of Fischer, Cudhill instead of Cundhill). Cardinale reference formatting amiss (633)

---

## [Editor Report]

There is a very thorough review of this resubmisison. The reviewer and handling editor agree on minor revisions. That seems the right decision.

---

## [Reviewer Report]

A nice job on revisions. Especially love hearing those participant voices! I caught a few typographical issues on this last read that you might want to fix up, but you could also manage it within the typesetting process.

Abstract Line 16 errant full stop after frequent

Line 57 something missing before EU, “like”?

67 “Risher” still here instead of Fischer

81-83 I’d say “that the provision of “training and capacity building opportunities for communities and practitioners” is one of six proposed actions to support the UN Decade on Ecosystem Restoration”

87, discrete, not discreet

117 I’d start a new paragraph around here to break this big one up

139 I’d say that instead of which (but that is a style preference as per Strunk and White, e.g., I go “which-hunting”)

Seem to be some resolution issues with Figure 2 the typesetter may query, depending on the original one uploaded. Other figures looking nice.

292-306 Some of the sentences still warrant periods not question marks. They are not phrased as questions. (#3, 7 and 8) Or rephrase as questions. Some are sentence fragments, as noted in last review.

316 start new paragraph when you go back to talking about data prep of interviews and surveys.

413 others, and complementary

528 Sand

557 I’d use participant singular here

574 full stop missing

581 remove “and”? Or some other fix.